# Towards Universal Visual Reward and Representation via Value-Implicit Pre-Training

## Abstract

We introduce **V**alue-**I**mplicit **P**re-training (VIP), a self-supervised pre-trained visual representation capable of generating dense and smooth reward functions for unseen robotic tasks. VIP casts representation learning from human videos as an *offline goal-conditioned reinforcement learning* problem and derives a self-supervised *dual* goal-conditioned value-function objective that does not depend on actions, enabling pre-training on unlabeled human videos. Theoretically, VIP can be understood as a novel *implicit* time contrastive learning that makes for temporally smooth embedding that enables the value function to be implicitly defined via the embedding distance, which can be used as the reward function for any downstream task specified through goal images. Trained on large-scale Ego4D human videos and without any fine-tuning on task-specific robot data, VIP's frozen representation can provide dense visual reward for an extensive set of simulated and **real-robot** tasks, enabling diverse reward-based policy learning methods, including visual trajectory optimization and online/offline RL, and significantly outperform all prior pre-trained representations. Notably, VIP can enable *few-shot* offline RL on a suite of real-world robot tasks with as few as 20 trajectories. Project website: https://sites.google.com/view/rl-vip

## 1 Value-Implicit Pre-Training

Due to space limit, we provide the full version of this section in Appendix D.

### 1.1 Foundation: Self-Supervised Value Learning from Human Videos

While human videos are out-of-domain data for robots, they are *in-domain* for learning a goal-conditioned human policy. Given that human videos naturally contain goal-directed behavior, one reasonable idea of utilizing offline human videos for representation learning is to solve an offline goal-conditioned RL problem over the space of human policies and then extract the learned visual representation. However, this idea is seemingly implausible because the offline human dataset does not come with any action labels that are typically required for *policy* learning. Our key insight is that, for a suitable choice of offline policy optimization problem, we can solve for the *dual* value learning problem that does not depend on any action label in the offline dataset. In particular, leveraging the idea of Fenchel duality (Rockafellar, 1970) from convex optimization, we have the following result:

**Proposition 1.1.** *Under assumption of deterministic transition dynamics, the dual optimization problem of* (11) *is*

$$\max_\phi \min_V \mathbb{E}_{p(g)} \left[ (1-\gamma)\mathbb{E}_{\mu_0(o;g)}[V(\phi(o);\phi(g))] + \log \mathbb{E}_{(o,o';g)\sim D} \left[ \exp\left(r(o,g) + \gamma V(\phi(o');\phi(g)) - V(\phi(o),\phi(g))\right)\right]\right],$$

(1)

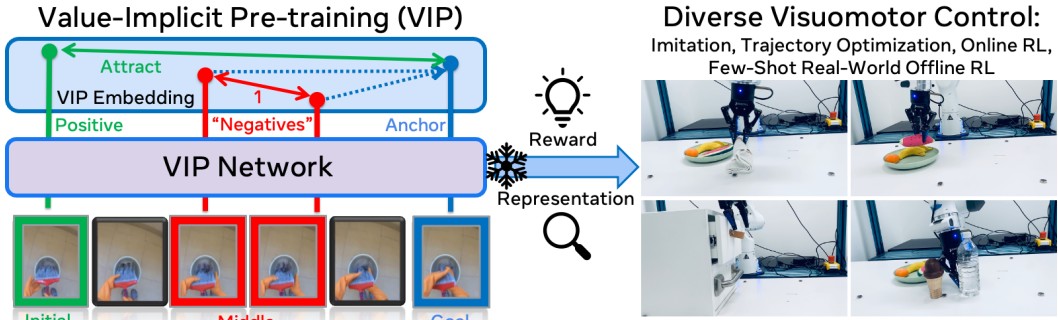

Figure 1: **Value-Implicit Pre-training (VIP)**. Pre-trained on large-scale, in-the-wild human videos, frozen VIP network can provide visual reward and representation for downstream robotics tasks and enable diverse visuomotor control strategies without any task-specific fine-tuning.

where $\mu_0(o; g)$ *is the goal-conditioned initial observation distribution, and* $D(o, o'; g)$ *is the goal-conditioned distribution of two consecutive observations in dataset* $D$.

As shown, actions do not appear in the objective. Furthermore, since all expectations in (12) can be sampled using the offline dataset, this dual value-function objective can be self-supervised with an appropriate choice of reward function. In particular, since our goal is to acquire a value function that extracts a general notion of goal-directed task progress from passive offline human videos, we set $r(o, g) = \mathbb{I}(o == g) - 1$, which we refer to as $\tilde{\delta}_g(o)$ in shorthand. This reward provides a constant negative reward when $o$ is not the provided goal $g$, and does not require any task-specific engineering. The resulting value function $V(\phi(o); \phi(g))$ captures the discounted total number of steps required to reach goal $g$ from observation $o$, and will objective will encourage learning visual features $\phi$ that are amenable to predicting the discounted temporal distance between two frames in a human video sequence. With enough size and diversity in the training dataset, we hypothesize that this value function can generalize to completely unseen (robot) domains.

## 1.2  Analysis: Implicit Time Contrastive Learning

In this section, we show that (1) can be understood as a novel *implicit* temporal contrastive representation learning that acquires temporally smooth embedding distance over video sequences, underpinning VIP's efficacy jointly as a visual representation and reward for downstream control.

Assuming that the optimal $V^*$ is found in (1), with a few algebraic manipulation steps (see Appendix E for a derivation), we can massage (13) into an expression that resembles the InfoNCE (Oord et al., 2018) time contrastive learning (Sermanet et al., 2018) (see Appendix B.2 for a definition and additional background) objective:

$$\min_\phi (1 - \gamma)\mathbb{E}_{p(g),\mu_0(o;g)}\left[-\log \frac{e^{V^*(\phi(o);\phi(g))}}{\mathbb{E}_{D(o,o';g)}\left[\exp\left(\tilde{\delta}_g(o)+\gamma V^*(\phi(o');\phi(g))-V^*(\phi(o),\phi(g))\right)\right]^{\frac{-1}{(1-\gamma)}}}\right]$$

(2)

In particular, $p(g)$ can be thought of the distribution of "anchor" observations, $\mu_0(s; g)$ the distribution of "positives" samples, and $D(o, o'; g)$ the distribution of "negatives" samples. Since the value function encodes negative discounted temporal distance, due to the recursive nature of value temporal-difference (TD), in order for the one-step TD error to be globally minimized along a video sequence, observations that are temporally farther away from the goal will naturally be repelled farther away in the representation space compared to observations that are nearby in time. Therefore, the repulsion of the negative observations is an *implicit*, emergent property from the optimization of (2), instead of an explicit constraint as in standard (time) contrastive learning. In Appendix D, we detail how this implicit time contrast mechanism gives rise to a temporally smooth visual representation that makes for effective zero-shot reward-specification.

## 1.3 Algorithm: Value-Implicit Pre-Training (VIP)

Recall that $V^*$ is assumed to be known for the derivation in Section 1.2, but in practice, its analytical form is rarely known. Now, given that $V^*$ plays the role of a distance measure in our implicit time contrastive learning framework, a simple and intuitive way to approximate $V^*$ in practice is to *implicitly* parameterize it to be a choice of distance measure. In this work, we choose the common choice of the negative $L_2$ distance used in prior work Sermanet et al. (2018); Nair et al. (2022): $V^*(\phi(o), \phi(g)) := -\|\phi(o) - \phi(g)\|_2$. Altogether, VIP training is illustrated in Alg. 2; it is simple and its core training loop can be implemented in fewer than 10 lines of PyTorch code (Alg. 3).

---

**Algorithm 1** Value-Implicit Pre-Training (VIP)

---

1: **Require**: Offline (human) videos $D = \{(o_1^i, ..., o_{i_h}^i)\}_{i=1}^N$, visual architecture $\phi$
2: **for** number of training iterations **do**
3:      Sample sub-trajectories $\{o_t^i, ..., o_k^i, o_{k+1}^i, ..., o_T^i\}_{i=1}^B \sim D, t \in [1, i_h - 1], t \le k < T, T \in (t, i_h], \forall i$
4:      $\mathcal{L}(\phi) := \frac{1-\gamma}{B} \sum_{i=1}^B \left[\|\phi(o_t^i) - \phi(o_T^i)\|_2\right] + \log \frac{1}{B} \sum_{i=1}^B \left[\exp\left(\|\phi(o_k^i) - \phi(o_T^i)\|_2 - \tilde{\delta}_{o_T^i}(o_k^i) - \gamma \|\phi(o_{k+1}^i) - \phi(o_T^i)\|_2\right)\right]$

5:      Update $\phi$ using SGD: $\phi \leftarrow \phi - \alpha_\phi \nabla\mathcal{L}(\phi)$

---

## 2 Experiments

In this section, we demonstrate VIP's effectiveness as both a pre-trained visual reward and representation on three distinct reward-based policy learning settings. Due to space limit, we delve into results directly, and all omitted experimental details are contained in App. G; additional results and analysis are presented in App.I. At a high level, VIP fixes the visual architecture (ResNet50) and pre-training dataset (Ego4D) as a state-of-art pre-trained representation R3M (Nair et al., 2022), differing pri-

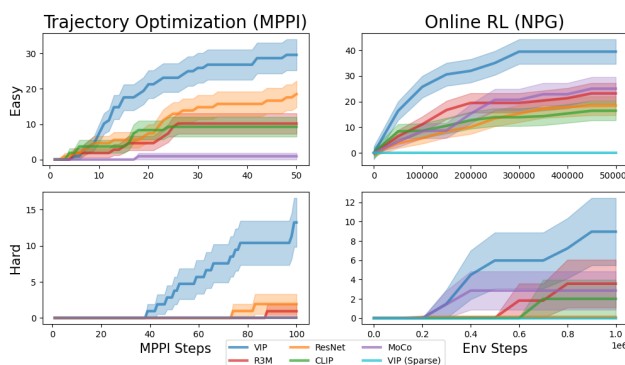

Figure 2: Visual traj. opt. and RL results (max success rate %).

marily in the training objective. We use FrankaKitchen (Gupta et al., 2019) for evaluation. Each task is specified via only a goal image, requiring the pre-trained representations to provide embedding-distance based reward (4) and visual encoding.

### 2.1 Trajectory Optimization & Online Reinforcement Learning

We evaluate pre-trained representations' capability as pure visual reward functions by using them to directly synthesize a sequence of actions using a standard trajectory optimization algorithm. We also evaluate online RL, which provides improved exploration but comes with the added challenge of demanding the pre-trained representation to provide both the visual reward and representation for learning a closed-loop policy. In Figure 2, we report each representation's cumulative success rate averaged over task configurations and random seeds (3 seeds * 3 cameras * 12 tasks = 108 runs).

Examining the MPPI results, we see that VIP is substantially better than all baselines in both Easy and Hard settings, and is the only representation that makes non-trivial progress on the Hard setting. These results demonstrate that VIP has superior capability as a pure visual reward function. In Fig. 3, we couple VIP and the strongest baselines (R3M, Resnet)'s with increasingly powerful MPPI optimizers (i.e., more trajectories per optimization step). As shown, while VIP steadily benefits from stronger optimizers and can reach an average success rate of **44%**, baselines often do *worse* when MPPI becomes more powerful, suggesting that their reward landscapes are filled with local minima that do not correlate with task progress and are easily exploited by (stronger) optimizers.

Switching gear to online RL, VIP again achieves consistently superior performance, demonstrating its joint effectiveness as visual reward and representation. VIP (Sparse)'s inability to solve any

Table 1: Real-robot offline RL results (success rate % averaged over 10 rollouts with standard deviation reported).

| Environment | Pre-Trained | | | | | In-Domain | |
| --- | --- | --- | --- | --- | --- | --- | --- |
| | VIP-RWR | VIP-BC | R3M-RWR | R3M-BC | Scratch-BC | VIP-RWR | VIP-BC |
| CloseDrawer | $\mathbf{100} \pm 0$ | $50 \pm 50$ | $80 \pm 40$ | $10 \pm 30$ | $30 \pm 46$ | $0 \pm 0$ | $0^* \pm 0$ |
| PushBottle | $\mathbf{90} \pm 30$ | $50 \pm 50$ | $70 \pm 46$ | $50 \pm 50$ | $40 \pm 48$ | $0^* \pm 0$ | $0^* \pm 0$ |
| PlaceMelon | $\mathbf{60} \pm 48$ | $10 \pm 30$ | $0 \pm 0$ | $0 \pm 0$ | $0 \pm 0$ | $0^* \pm 0$ | $0^* \pm 0$ |
| FoldTowel | $\mathbf{90} \pm 30$ | $20 \pm 40$ | $0 \pm 0$ | $0 \pm 0$ | $0 \pm 0$ | $0^* \pm 0$ | $0^* \pm 0$ |

task indicates the necessity of dense reward in solving these challenging visual manipulation tasks. Whereas sparse reward still requires human engineering via installing additional sensors (Rajeswar et al., 2021; Singh et al., 2019) and faces exploration challenges (Nair et al., 2018), with VIP, the end-user has to provide only a goal image, and, without any additional state or reward instrumentation, can expect a significant improvement in performance.

## 2.2 Real-World Few-Shot Offline Reinforcement Learning

Finally, we demonstrate how VIP's reward and representation can power a simple and practical system for real-world robot learning in the form of *few-shot* offline reinforcement learning, making offline RL simple, sample-efficient, and more effective than BC with almost no added complexity.

To this end, we consider a simple reward-weighted regression (RWR) (Peters & Schaal, 2007; Peng et al., 2019) approach, in which the reward and the encoder are provided by the pre-trained model $\phi$:

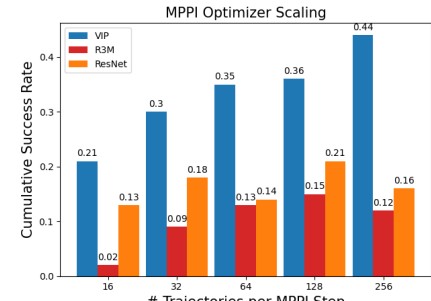

Figure 3: VIP benefits from scaling compute for downstream trajectory optimization.

$$\mathcal{L}(\pi) = -\mathbb{E}_{D_{\text{task}}} \left[ \exp(\tau \cdot R(o, o'; \phi, g)) \log \pi(a \mid \phi(o)) \right],$$
(3)

where $R$ is defined via (4) and $\tau$ is the temperature scale. Compared to BC, which would be (3) with uniform weights to all transitions, RWR can focus policy learning on transitions that have high rewards (i.e., high task progress) under the deployed representation.

We introduce 4 tabletop manipulation tasks (see Figure 1 and Figure 10) requiring a real 7-DOF Franka robot to manipulate objects drawn from distinct categories of objects. For each task, we collect in-domain, task-specific offline data $D_{\text{task}}$ of $\sim 20$ demonstrations with randomized object initial placements for policy learning; we provide detailed task and experiment descriptions in Appendix H.

The average success rate (%) and standard deviation across 10 test rollouts are reported in Table 1. As shown, VIP-RWR improves upon VIP-BC on all tasks and provides substantial benefit in the harder tasks that are multi-stage in nature. In contrast, R3M-RWR, while able to improve R3M-BC on the simpler two tasks involving pushing an object, fails to make any progress on the harder tasks. The low performance of BC-based methods on the harder PickPlaceMelon and FoldTowel tasks indicates that in this low-data regime, regardless of the quality of visual representation, good reward information is necessary for task success. Finally, *in-domain* methods all fail in this low-data regime. Altogether, these results corroborate the necessity of pre-training in achieving real-world few-shot offline RL and highlight the unique effectiveness of VIP in realizing this goal.

## 3 Conclusion

We have proposed Value-Implicit Pre-training (VIP), a self-supervised value-based pre-training objective that is highly effective in providing both the visual reward and representation for downstream unseen robotics tasks. VIP is derived from first principles of dual reinforcement learning and admits an appealing connection to an implicit and more powerful formulation of time contrastive learning, which captures long-range temporal dependency and injects local temporal smoothness in the representation to make for effective zero-shot reward specification. Trained entirely on diverse, in-the-wild human videos, VIP demonstrates significant gains over state-of-art pre-trained representations on an extensive set of policy learning settings. Notably, VIP can enable simple and sample-efficient real-world offline RL with just handful of trajectories. Altogether, we believe that VIP makes an important contribution in both the algorithmic frontier of visual pre-training for RL and practical real-world robot learning.

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

# Part I

# Appendix

## Table of Contents

## A  Problem Setting and Background

In this section, we describe our problem setting of out-of-domain pre-training and provide formalism for downstream representation evaluation. Additional background on goal-conditioned reinforcement learning and contrastive learning is included in Appendix B.

### A.1  Out-of-Domain Pre-Training Visual Representation

We consider the problem setting of pre-training a frozen visual encoder for downstream control tasks (Shah & Kumar, 2021; Parisi et al., 2022; Nair et al., 2022). More specifically, we have access to a training set of video data $D = \{v_i := (o_1^i, ..., o_{i_h}^i)\}_{i=1}^N$, where each $o \in \mathbb{R}^{H \times W \times 3}$ is a raw RGB image; note that this formalism also captures standard image datasets (e.g., ImageNet), if we take $i_h = 1$ for all $v_i$. Like prior works, we assume $D$ to be out-of-domain and does not include any robot task or domain-specific data. A learning algorithm $\mathcal{A}$ ingests this training data and outputs a visual encoder $\phi := \mathcal{A}(D) : \mathbb{R}^{H \times W \times 3} \to K$, where $K$ is the embedding dimension.

### A.2  Representation Evaluation

Given a choice of representation $\phi$, every evaluation task can be instantiated as a Markov decision process $\mathcal{M}(\phi) := (\phi(O), A, R(o_t, o_{t+1}; \phi, g), T, \gamma, g)$, in which the state space is the induced space of observation embeddings, and the task is specified via a (set of) goal image(s) $g$. Specifically, for a given transition tuple $(o_t, o_{t+1})$, we define the reward to be the goal-embedding distance difference (Lee et al., 2021; Li et al., 2022):

$$R(o_t, o_{t+1}; \phi, \{g\}) := \mathcal{S}_\phi(o_{t+1}; g) - \mathcal{S}_\phi(o_t; g) := (1-\gamma)\mathcal{S}_\phi(o_{t+1}; g) + (\gamma\mathcal{S}_\phi(o_{t+1}; g) - \mathcal{S}_\phi(o_t; g)), \tag{4}$$

where $\mathcal{S}_\phi$ is a choice of distance function in the $\phi$-representation space; in this work, we set $\mathcal{S}_\phi(o_t; g) := -\|\phi(o_t) - \phi(g)\|_2$. This reward function can be interpreted as a raw embedding distance reward with a reward shaping (Ng et al., 1999) term that encourages making progress towards the goal. This preserves optimal policy but enables more efficient and robust policy learning.

Under this formalism, parameters of $\phi$ are frozen during policy learning (it is considered a part of the MDP), and we want to learn a policy $\pi : \mathbb{R}^K \to A$ that outputs an action based on the embedded observation $a \sim \pi(\phi(o))$.

## B  Additional Background

### B.1  Goal-Conditioned Reinforcement Learning

This section is adapted from Ma et al. (2022b). We consider a goal-conditioned Markov decision process from visual state space: $\mathcal{M} = (O, A, G, r, T, \mu_0, \gamma)$ with state space $O$, action space $A$, reward $r(o, g)$, transition function $o' \sim T(o, a)$, the goal distribution $p(g)$, and the goal-conditioned initial state distribution $\mu_0(o; g)$, and discount factor $\gamma \in (0, 1]$. We assume the state space $O$ and the goal space $G$ to be defined over RGB images. The objective of goal-conditioned RL is to find a goal-conditioned policy $\pi : O \times G \to \Delta(A)$ that maximizes the discounted cumulative return:

$$J(\pi) := \mathbb{E}_{p(g), \mu_0(o;g), \pi(a_t|s_t, g), T(o_{t+1}, |o_t, a_t)} \left[ \sum_{t=0}^\infty \gamma^t r(o_t; g) \right] \tag{5}$$

The *goal-conditioned* state-action occupancy distribution $d^\pi(o, a; g) : O \times A \times G \to [0, 1]$ of $\pi$ is

$$d^\pi(o, a; g) := (1 - \gamma) \sum_{t=0}^\infty \gamma^t \Pr(o_t = o, a_t = a \mid o_0 \sim \mu_0(o; g), a_t \sim \pi(o_t; g), o_{t+1} \sim T(o_t, a_t)) \tag{6}$$

which captures the goal-conditioned visitation frequency of state-action pairs for policy $\pi$. The state-occupancy distribution then marginalizes over actions: $d^\pi(o; g) = \sum_a d^\pi(o, a; g)$. Then, it

follows that $\pi(a \mid o, g) = \frac{d^\pi(o,a;g)}{d^\pi(o;g)}$. A state-action occupancy distribution must satisfy the *Bellman flow constraint* in order for it to be an occupancy distribution for some stationary policy $\pi$:

$$\sum_a d(o, a; g) = (1 - \gamma)\mu_0(o; g) + \gamma \sum_{\tilde{o}, \tilde{a}} T(s \mid \tilde{o}, \tilde{a}) d(\tilde{o}, \tilde{a}; g), \qquad \forall o \in O, g \in G \qquad (7)$$

We write $d^\pi(o, g) = p(g)d^\pi(o; g)$ as the joint goal-state density induced by $p(g)$ and the policy $\pi$. Finally, given $d^\pi$, we can express the objective function (5) as $J(\pi) = \frac{1}{1-\gamma}\mathbb{E}_{(o,g)\sim d^\pi(o,g)}[r(o, g)]$.

## B.2 InfoNCE & Time Contrastive Learning.

As VIP can be understood as a implicit and smooth time contrastive learning objective, we provide additional background on the InfoNCE Oord et al. (2018) and time contrastive learning (TCN) (Sermanet et al., 2018) objective to aid comparison in Section D.2.

InfoNCE is an unsupervised contrastive learning objective built on the noise contrastive estimation (Gutmann & Hyvärinen, 2010) principle. In particular, given an "anchor" datum $x$ (otherwise known as context), and distribution of positives $x_{\text{pos}}$ and negatives $x_{\text{neg}}$, the InfoNCE objective optimizes

$$\min_\phi \mathbb{E}_{x_{\text{pos}}} \left[ -\log \frac{\mathcal{S}_\phi(x, x_{\text{pos}})}{\mathbb{E}_{x_{\text{neg}}} \mathcal{S}_\phi(x, x_{\text{neg}})} \right], \qquad (8)$$

where $\mathbb{E}_{x_{\text{neg}}}$ is often approximated with a fixed number of negatives in practice.

It is shown in Oord et al. (2018) that optimizing (8) is maximizing a lower bound on the mutual information $\mathcal{I}(x, x_{\text{pos}})$, where, with slight abuse of notation, $x$ and $x_{\text{pos}}$ are interpreted as random variables.

TCN is a contrastive learning objective that learns a representation that in timeseries data (e.g., video trajectories). The original work (Sermanet et al., 2018) considers multi-view videos and perform contrastive learning over frames in separate videos; in this work, we consider the single-view variant. At a high level, TCN attracts representations of frames that are temporally close, while pushing apart those of frames that are farther apart in time. More precisely, given three frames sampled from a video sequence $(o_{t_1}, o_{t_2}, o_{t_3})$, where $t_1 < t_2 < t_3$, TCN would attract the representations of $o_{t_1}$ and $o_{t_2}$ and repel the representation of $o_{t_3}$ from $o_{t_1}$. This idea can be formally expressed via the following objective:

$$\min_\phi \mathbb{E}_{(o_{t_1}, o_{t_2} > t_1) \sim D} \left[ -\log \frac{\mathcal{S}_\phi(o_{t_1}; o_{t_2})}{\mathbb{E}_{o_{t_3} \mid t_3 > t_2 \sim D} [\mathcal{S}_\phi(o_{t_1}; o_{t_3})]} \right] \qquad (9)$$

Given a "positive" window of $K$ steps and a uniform distribution among valid positive samples, we can write (9) as

$$\min_\phi \frac{1}{K} \sum_{k=1}^{K} \mathbb{E}_{(o_{t_1}, o_{t_1+k}) \sim D} \left[ -\log \frac{\mathcal{S}_\phi(o_{t_1}; o_{t_1+k})}{\mathbb{E}_{o_{t_3} \mid t_3 > t_1+k \sim D} [\mathcal{S}_\phi(o_{t_1}; o_{t_3})]} \right], \qquad (10)$$

in which each term inside the expectation is a standalone InfoNCE objective tailored to observation sequence data.

## C  Related Work

We review relevant literature on (1) Out-of-Domain Representation Pre-Training for Control, (2) Perceptual Reward Learning from Human Videos, and (3) Goal-Conditioned RL as Representation Learning.

**Out-of-Domain Representation Pre-Training for Control.** Bootstrapping visual control using frozen representations learned pre-trained on out-of-domain non-robot data is a nascent field that has seen fast progress over the past year. Shah & Kumar (2021) demonstrates that pre-trained ResNet (He et al., 2016) representation on ImageNet (Deng et al., 2009) serves as effective visual backbone

for simulated dexterous manipulation RL tasks. Parisi et al. (2022) finds ResNet models trained with unsupervised objectives, such as momentum contrastive learning (MOCO) (He et al., 2020), to surpass supervised objectives (e.g, image classification) for both visual navigation and control tasks. Xiao et al. (2022) pre-trains visual representation on human video data (Goyal et al., 2017; Shan et al., 2020) using masked-autoencoding (He et al., 2022). Along this axis, the closest work to ours is Nair et al. (2022), which is also pre-trained on the Ego4D dataset and attempts to capture temporal information in the videos by using time-contrastive learning (Sermanet et al., 2018); it additionally leverages textual descriptions associated with the videos to encode semantic information. In contrast, our objective is fully self-supervised without dependence on textual annotations. Furthermore, VIP is the first to propose using a RL-based objective for out-of-domain pre-training and is capable of producing generalizable dense reward signals.

**Perceptual Reward Learning from Human Videos.** Human videos provide a rich natural source of reward and representation learning for robotic learning. Most prior works exploit the idea of learning an invariant representation between human and robot domains to transfer the demonstrated skills (Sermanet et al., 2016, 2018; Schmeckpeper et al., 2020; Chen et al., 2021; Xiong et al., 2021; Zakka et al., 2022; Bahl et al., 2022). However, training these representations require task-specific human *demonstration* videos paired with robot videos solving the same task, and cannot leverage the large amount of "in-the-wild" human videos readily available. As such, these methods require robot data for training, and learn rewards that are task-specific and do not generalize beyond the tasks they are trained on. In contrast, VIP do not make any assumption on the quality or the task-specificity of human videos and instead pre-trains an (implicit) value function that aims to capture task-agnostic goal-oriented progress, which can generalize to completely unseen robot domains and tasks.

**Goal-Conditioned RL as Representation Learning.** Our pre-training method is also related to the idea of treating goal-conditioned RL as representation learning. Chebotar et al. (2021) shows that a goal-conditioned Q-function trained with offline in-domain multi-task robot data learns an useful visual representation that can accelerate learning for a new downstream task in the same domain. Eysenbach et al. (2022) shows that goal-conditioned Q-learning with a particular choice of reward function can be understood as performing contrastive learning. In contrast, our theory introduces a new implicit time contrastive learning, and states that for *any* choice of reward function, the dual formulation of a regularized offline GCRL objective can be cast as implicit time contrast. This conceptual bridge also explains why VIP's learned embedding distance is temporally smooth and can be used as an universal reward mechanism. Finally, whereas these two works are limited to training on in-domain data with robot action labels, VIP is able to leverage diverse out-of-domain human data for visual representation pre-training, overcoming the inherent limitation of robot data scarcity for in-domain training.

Our work is also closely related to Ma et al. (2022b), which first introduced the dual offline GCRL objective based on Fenchel duality (Rockafellar, 1970; Nachum & Dai, 2020; Ma et al., 2022a). Whereas Ma et al. (2022b) assumes access to the true state information and focuses on the offline GCRL setting using in-domain offline data with robot action labels, we extend the dual objective to enable out-of-domain, action-free pre-training from human videos. Our particular dual objective also admits a novel implicit time contrastive learning interpretation, which simplifies VIP's practical implementation by letting the value function be implicitly defined instead of a deep neural network as in Ma et al. (2022b).

# D    Value-Implicit Pre-Training (Full-Version)

In this section, we demonstrate how a self-supervised value-function objective can be derived from computing the dual of an offline RL objective on passive human videos (Section D.1). Then, we show how this objective amounts to a novel implicit formulation of temporal contrastive learning (Section D.2), which naturally lends a temporally and locally smooth embedding favorable for downstream visual reward specification. Finally, we leverage this contrastive interpretation to instantiate a simple implementation (<10 lines of PyTorch code) of our dual value objective that does

452 not explicitly learn a value network (Section D.3), culminating in our final algorithm, Value-Implicit
453 Pre-training (VIP).

## D.1 Foundation: Self-Supervised Value Learning from Human Videos

455 While human videos are out-of-domain data for robots, they are *in-domain* for learning a goal-
456 conditioned policy $\pi_H$ over human actions, $a^H \sim \pi^H(\phi(o) \mid \phi(g))$, for some human action space
457 $A^H$. Therefore, given that human videos naturally contain goal-directed behavior, one reasonable idea
458 of utilizing offline human videos for representation learning is to solve an offline goal-conditioned
459 RL problem over the space of human policies and then extract the learned visual representation. To
460 this end, we consider the following KL-regularized offline RL objective (Nachum et al., 2019) for
461 some to-be-specified reward $r(o, g)$:

$$\max_{\pi_H, \phi} \mathbb{E}_{\pi^H} \left[ \sum_t \gamma^t r(o; g) \right] - (d^{\pi_H}(o, a^H; g) \| d^D(o, \tilde{a}^H; g)), \tag{11}$$

462 where $d^{\pi_H}(o, a^H; g)$ is the distribution over observations and actions $\pi_H$ visits conditioned on $g$.
463 Observe that a "dummy" action $\tilde{a}$ is added to every transition $(o_h^i, \tilde{a}_h^i, o_{h+1}^i)$ in the dataset $D$ so that
464 the KL regularization is well-defined, and $\tilde{a}_i^h$ can be thought of as the unobserved *true* human action
465 taken to transition from observation $o_h^i$ to $o_{h+1}^i$. While this objective is mathematically sound and
466 encourages learning a conservative $\pi^H$, it is seemingly implausible because the offline dataset $D^H$
467 does not come with any action labels nor can $A^H$ be concretely defined in practice. However, what
468 this objective does provide is an elegant *dual* objective over a value function that does not depend on
469 any action label in the offline dataset. In particular, leveraging the idea of Fenchel duality (Rockafellar,
470 1970) from convex optimization, we have the following result:

471 **Proposition D.1.** *Under assumption of deterministic transition dynamics, the dual optimization*
472 *problem of* (11) *is*

473 $$\max_\phi \min_V \mathbb{E}_{p(g)} \left[ (1 - \gamma) \mathbb{E}_{\mu_0(o; g)} [V(\phi(o); \phi(g))] + \log \mathbb{E}_{(o, o'; g) \sim D} \left[ \exp \left( r(o, g) + \gamma V(\phi(o'); \phi(g)) - V(\phi(o), \phi(g)) \right) \right] \right], \tag{12}$$

474 *where $\mu_0(o; g)$ is the goal-conditioned initial observation distribution, and $D(o, o'; g)$ is the goal-*
475 *conditioned distribution of two consecutive observations in dataset $D$.*

476 As shown, actions do not appear in the objective. Furthermore, since all expectations in (12) can be
477 sampled using the offline dataset, this dual value-function objective can be self-supervised with an
478 appropriate choice of reward function. In particular, since our goal is to acquire a value function that
479 extracts a general notion of goal-directed task progress from passive offline human videos, we set
480 $r(o, g) = \mathbb{I}(o == g) - 1$, which we refer to as $\tilde{\delta}_g(o)$ in shorthand. This reward provides a constant
481 negative reward when $o$ is not the provided goal $g$, and does not require any task-specific engineering.
482 The resulting value function $V(\phi(o); \phi(g))$ captures the discounted total number of steps required to
483 reach goal $g$ from observation $o$. Consequently, the overall objective will encourage learning visual
484 features $\phi$ that are amenable to predicting the discounted temporal distance between two frames in a
485 human video sequence. With enough size and diversity in the training dataset, we hypothesize that
486 this value function can generalize to completely unseen (robot) domains and tasks.

## D.2 Analysis: Implicit Time Contrastive Learning

488 While (12) will learn some useful visual representation via temporal value function optimization,
489 in this section, we show that it can be understood as a novel *implicit* temporal contrastive learning
490 objective that acquires temporally smooth embedding distance over video sequences, underpinning
491 VIP's efficacy jointly as a visual representation and reward for downstream control.

492 We begin by simplifying the expression in (12) by first assuming that the optimal $V^*$ is found:

493 $$\min_\phi \mathbb{E}_{p(g)} \left[ (1 - \gamma) \mathbb{E}_{\mu_0(o; g)} [-V^*(\phi(o); \phi(g))] + \log \mathbb{E}_{D(o, o'; g)} \left[ \exp \left( \tilde{\delta}_g(o) + \gamma V(\phi(o'); \phi(g)) - V(\phi(o), \phi(g)) \right) \right]^{-1} \right], \tag{13}$$

494 where we have also re-written the maximization problem as a minimization problem. Now, after
495 few algebraic manipulation steps (see App. E for a derivation), if we think of $V^*(\phi(o); \phi(g))$ as a

*similarity metric* in the embedding space, then we can massage (13) into an expression that resembles the InfoNCE (Oord et al., 2018) time contrastive learning (Sermanet et al., 2018) (see App. B.2 for a definition and additional background) objective:

$$\min_\phi (1-\gamma)\mathbb{E}_{p(g),\mu_0(o;g)}\left[-\log \frac{e^{V^*(\phi(o);\phi(g))}}{\mathbb{E}_{D(o,o';g)}\left[\exp\left(\tilde{\delta}_g(o)+\gamma V^*(\phi(o');\phi(g))-V^*(\phi(o),\phi(g))\right)\right]^{\frac{-1}{(1-\gamma)}}}\right] \quad (14)$$

In particular, $p(g)$ can be thought of the distribution of "anchor" observations, $\mu_0(s;g)$ the distribution of "positive" samples, and $D(o,o';g)$ the distribution of "negative" samples. Counter-intuitively and in contrast to standard single-view time contrastive learning (TCN), in which the positive observations are temporally closer to the anchor observation than the negatives, (14) has the positives to be as temporally far away as possible, namely the initial frame in the the same video sequence, and the negatives to be middle frames sampled in between. This departure is accompanied by the equally intriguing deviation of the lack of explicit repulsion of the negatives from the anchor; instead, they are simply encouraged to minimize the (exponentiated) one-step temporal-difference error in the representation space (the denominator in (14)); see Fig. 1. Now, since the value function encodes negative discounted temporal distance, due to the recursive nature of value temporal-difference (TD), in order for the one-step TD error to be globally minimized along a video sequence, observations that are temporally farther away from the goal will naturally be repelled farther away in the representation space compared to observations that are nearby in time; in App. E.3, we formalize this intuition and show that this repulsion always holds for optimal paths. Therefore, the repulsion of the negative observations is an *implicit*, emergent property from the optimization of (14), instead of an explicit constraint as in standard (time) contrastive learning.

Now, we dive into why this *implicit* time contrastive learning is desirable. First, the explicit attraction of the initial and goal frames enables capturing *long-range* semantic temporal dependency as two frames that meaningfully indicate the beginning and end of a task are made close in the embedding space. This closeness is also well-defined due to the one-step TD backup that makes every embedding distance recursively defined to be the discounted number of timesteps to the goal frame. Combined with the implicit yet structured repulsion of intermediate frames,

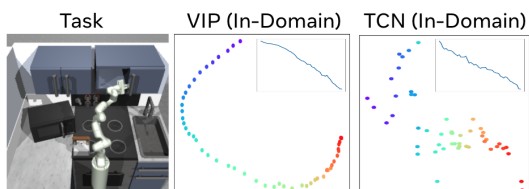

Figure 4: Learned 2D representation of a held-out task demonstration by VIP and TCN trained on task-specific in-domain data. The color gradient indicates trajectory time progression (purple for beginning, red for end). The inset plots are embedding distances to last frame.

this push-and-pull mechanism helps inducing a *temporally smooth* and consistent representation. In particular, as we pass a video sequence in the training set through the trained representation, the embedding should be structured such that two trends emerge: (1) neighboring frames are close-by in the embedding space, (2) their distances to the last (goal) frame smoothly decrease due to the recursively defined embedding distances. To validate this intuition, in Fig. 4, we provide a simple toy example comparing implicit vs. standard time contrastive learning when trained on *in-domain, task-specific* demonstrations; details are included in App. G.2. As shown, standard time contrastive learning only enforces a coarse notion of temporal consistency and learns a non-locally smooth representation that exhibits many local minima. In contrast, VIP learns a much better structured embedding that is indeed temporally consistent and locally smooth. As we will show, the prevalence of sharp "bumps" in the embedding distance as in TCN can be easily exploited by the control algorithm, and VIP's ability to generate long-range temporally smooth embedding is the key ingredient for its effective downstream zero-shot reward-specification.

### D.3 Algorithm: Value-Implicit Pre-Training (VIP)

The theoretical development in the previous two sections culminates in *Value Implicit Pre-Training* (VIP), a simple value-based self-supervised pre-training objective, in which the value function is implicitly represented via the learned embedding distance.

Recall that $V^*$ is assumed to be known for the derivation in Section D.2, but in practice, its analytical form is rarely known. Now, given that $V^*$ plays the role of a distance measure in our implicit time contrastive learning framework, a simple and practical way to approximate $V^*$ is to simply set it to be a choice of similarity metric, bypassing having to explicitly parameterize it as a neural network. In this work, we choose the common choice of the negative $L_2$ distance used in prior work Sermanet et al. (2018); Nair et al. (2022): $V^*(\phi(o), \phi(g)) := - \|\phi(o) - \phi(g)\|_2$. Given this choice, our final representation learning objective is as follows:

$$\mathrm{L}(\phi) = \mathbb{E}_{p(g)} \left[ (1 - \gamma) \mathbb{E}_{\mu_0(o;g)} \left[ \|\phi(o) - \phi(g)\|_2 \right] + \log \mathbb{E}_{(o,o';g) \sim D} \left[ \exp \left( \|\phi(o) - \phi(g)\|_2 - \tilde{\delta}_g(o) - \gamma \|\phi(o') - \phi(g)\|_2 \right) \right] \right], (15)$$

in which we also absorb the exponent of the log-sum-exp term in 13 into the inner $\exp(\cdot)$ term via an Jensen's inequality; we found this upper bound to be numerically more stable. To sample video trajectories from $D$, because any sub-trajectory of a video is also a valid video sequence, VIP samples these sub-trajectories and treats their initial and last frames as samples from the goal and initial-state distributions (Step 3 in Alg. 2). Altogether, VIP training is illustrated in Alg. 2; it is simple and its core training loop can be implemented in fewer than 10 lines of PyTorch code (Alg. 3 in App. F.3).

---

**Algorithm 2** Value-Implicit Pre-Training (VIP)

1: **Require**: Offline (human) videos $D = \{(o_1^i, ..., o_{h_i}^i)\}_{i=1}^N$, visual architecture $\phi$
2: **for** number of training iterations **do**
3:     Sample sub-trajectories $\{o_t^i, ..., o_k^i, o_{k+1}^i, ..., o_T^i\}_{i=1}^B \sim D, t \in [1, h_i - 1], t \le k < T, T \in (t, h_i], \forall i$
4:     $\mathcal{L}(\phi) := \frac{1-\gamma}{B} \sum_{i=1}^B \left[ \|\phi(o_t^i) - \phi(o_T^i)\|_2 \right] + \log \frac{1}{B} \sum_{i=1}^B \left[ \exp \left( \|\phi(o_k^i) - \phi(o_T^i)\|_2 - \tilde{\delta}_{o_T^i}(o_k^i) - \gamma \|\phi(o_{k+1}^i) - \phi(o_T^i)\|_2 \right) \right]$
5:     Update $\phi$ using SGD: $\phi \leftarrow \phi - \alpha_\phi \nabla \mathcal{L}(\phi)$

---

# E    Technical Derivations and Proofs

## E.1    Proof of Proposition D.1

We first reproduce Proposition D.1 for ease of reference:

**Proposition E.1.** *Under assumption of deterministic transition dynamics, the dual optimization problem of*

$$\max_{\pi_H, \phi} \mathbb{E}_{\pi^H} \left[ \sum_t \gamma^t r(o; g) \right] - (d^{\pi^H}(o, a^H; g) \| d^D(o, \tilde{a}^H; g)), \tag{16}$$

*is*

$$\max_\phi \min_V \mathbb{E}_{p(g)} \left[ (1 - \gamma) \mathbb{E}_{\mu_0(o;g)} [V(\phi(o); \phi(g))] + \log \mathbb{E}_{D(o,o';g)} \left[ \exp \left( r(o, g) + \gamma V(\phi(o'); \phi(g)) - V(\phi(o), \phi(g)) \right) \right] \right],$$
$$\tag{17}$$

*where $\mu_0(o; g)$ is the goal-conditioned initial observation distribution, and $D(o, o'; g)$ is the goal-conditioned distribution of two consecutive observations in dataset $D$.*

*Proof.* We begin by rewriting (16) as an optimization problem over valid state-occupancy distributions. To this end, we have[1]

$$\max_\phi \max_{d(\phi(o), a; \phi(g)) \ge 0} \mathbb{E}_{d(\phi(o), \phi(g))} [r(o; g)] - (d(\phi(o), a; \phi(g)) \| d^D(\phi(o), \tilde{a}; \phi(g)))$$

(P)    s.t.    $\sum_a d(\phi(o), a; \phi(g)) = (1 - \gamma) \mu_0(o; g) + \gamma \sum_{\tilde{o}, \tilde{a}} T(o \mid \tilde{o}, \tilde{a}) d(\phi(\tilde{o}), \tilde{a}; \phi(g)), \forall o \in O, g \in G$

$$\tag{18}$$

Fixing a choice of $\phi$, the inner optimization problem operates over a $\phi$-induced state and goal space, giving us (18). Then, applying Proposition 4.2 of Ma et al. (2022b) to the inner optimization problem,

---

[1]We omit the human action superscript $H$ in this derivation.

we immediately obtain

$$\max_\phi \min_V \mathbb{E}_{p(g)}\big[(1-\gamma)\mathbb{E}_{\mu_0(o;g)}[V(\phi(o);\phi(g))]$$

$$\text{(D)} \qquad + \log \mathbb{E}_{d^D(\phi(o),a;\phi(g))}\big[\exp\big(r(o,g) + \gamma\mathbb{E}_{T(o'|o,a)}[V(\phi(o');\phi(g))] - V(\phi(o),\phi(g)))\big]\big]\tag{19}$$

Now, given our assumption that the transition dynamics is deterministic, we can replace the inner expectation $\mathbb{E}_{T(o'|o,a)}$ with just the observed sample in the offline dataset and obtain:

$$\max_\phi \min_V \mathbb{E}_{p(g)}\big[(1-\gamma)\mathbb{E}_{\mu_0(o;g)}[V(\phi(o);\phi(g))]$$

$$+ \log \mathbb{E}_{d^D(\phi(o),\phi(o');\phi(g))}\big[\exp\big(r(o,g) + \gamma V(\phi(o');\phi(g)) - V(\phi(o),\phi(g)))\big]\big]\tag{20}$$

Finally, sampling embedded states from $d^D(\phi(o),\phi(o');\phi(g))$ is equivalent to sampling from $D(o,o';g)$, assuming there is no embedding collision (i.e., $\phi(o) \neq \phi(o'), \forall o \neq o'$), which can be satisfied by simply augmenting any $\phi$ by concatenating the input to the end. Then, we have our desired expression:

$$\max_\phi \min_V \mathbb{E}_{p(g)}\big[(1-\gamma)\mathbb{E}_{\mu_0(o;g)}[V(\phi(o);\phi(g))] + \log \mathbb{E}_{D(o,o';g)}\big[\exp\big(r(o,g) + \gamma V(\phi(o');\phi(g)) - V(\phi(o),\phi(g)))\big]\big]\tag{21}$$

$\square$

## E.2 VIP Implicit Time Contrast Learning Derivation

This section provides all intermediate steps to go from (13) to (14). First, we have

$$\min_\phi \mathbb{E}_{p(g)}\left[(1-\gamma)\mathbb{E}_{\mu_0(o;g)}[-V^*(\phi(o);\phi(g))] + \log \mathbb{E}_{D(o,o';g)}\left[\exp\left(\tilde\delta_g(o) + \gamma V(\phi(o');\phi(g)) - V(\phi(o),\phi(g))\right)\right]^{-1}\right].\tag{22}$$

We can equivalently write this objective as

$$\min_\phi \mathbb{E}_{p(g)}\left[(1-\gamma)\mathbb{E}_{\mu_0(o;g)}[-\log e^{V^*(\phi(o);\phi(g))}] + \log \mathbb{E}_{D(o,o';g)}\left[\exp\left(\tilde\delta_g(o) + \gamma V(\phi(o');\phi(g)) - V(\phi(o),\phi(g))\right)\right]^{-1}\right].\tag{23}$$

Then,

$$\min_\phi \mathbb{E}_{p(g)}\left[(1-\gamma)\mathbb{E}_{\mu_0(o;g)}\left[-\log e^{V^*(\phi(o);\phi(g))} - \log \mathbb{E}_{D(o,o';g)}\left[\exp\left(\tilde\delta_g(o) + \gamma V(\phi(o');\phi(g)) - V(\phi(o),\phi(g))\right)\right]^{\frac{-1}{1-\gamma}}\right]\right]$$

$$= \min_\phi (1-\gamma)\mathbb{E}_{p(g),\mu_0(o;g)}\left[\log \frac{e^{-V^*(\phi(o);\phi(g))}}{\mathbb{E}_{D(o,o';g)}\left[\exp\left(\tilde\delta_g(o) + \gamma V(\phi(o');\phi(g)) - V(\phi(o),\phi(g))\right)\right]^{\frac{-1}{1-\gamma}}}\right].\tag{24}$$

This is (14) in the main text.

## E.3 VIP Implicit Repulsion

In this section, we formalize the implicit repulsion property of VIP objective ((14)); in particular, we prove that under certain assumptions, it always holds for optimal paths.

**Proposition E.2.** *Suppose $V^*(s;g) := -\|\phi(s) - \phi(g)\|_2$ for some $\phi$, under the assumption of deterministic dynamics (as in Proposition D.1), for any pair of consecutive states reached by the optimal policy, $(s_t, s_{t+1}) \sim \pi^*$, we have that*

$$\|\phi(s_t) - \phi(g)\|_2 > \|\phi(s_{t+1}) - \phi(g)\|_2,\tag{25}$$

*Proof.* First, we note that

$$V^*(s;g) = \max_a Q^*(s,a;g)\tag{26}$$

A proof can be found in Section 1.1.3 of Agarwal et al. (2019). Then, due to the Bellman optimality equation, we have that

$$Q^*(s,a;g) = r(s,g) + \gamma\mathbb{E}_{s'\sim T(s,a)}\max_{a'} Q^*(s',a';g)\tag{27}$$

Given that the dynamics is deterministic and (26), we have that

$$Q^*(s, a; g) = r(s, g) + \gamma V^*(s'; g) \tag{28}$$

Now, for $(s_t, a_t, s_{t+1}) \sim \pi^*$, this further simplifies to

$$V^*(s_t; g) = r(s_t, g) + \gamma V^*(s_{t+1}; g) \tag{29}$$

Note that since $V^*$ is also the optimal value function, given that $r(s_t, g) = \mathbb{I}(s_t = g) - 1$, $V^*(s_t; g)$ is the *negative* discounted distance of the shortest path between $s_t$ ans $g$. In particular, since $V^*(g; g) = 0$ by construction, we have that $V^*(s_t; g) = -\sum_{k=0}^{K} \gamma^k$ (this also clearly satisfies (29)), where the shortest path (i.e., the path $\pi^*$ takes) between $s_t$ and $g$ are $K$ steps long. Now, giving that we assume $V^*(s_t; g)$ can be expressed as $-\|\phi(s_t) - \phi(g)\|_2$ for some $\phi$, it immediately follows that

$$\|\phi(s_t) - \phi(g)\|_2 > \|\phi(s_{t+1}) - \phi(g)\|_2, \quad \forall (s_t, s_{t+1}) \sim \pi^* \tag{30}$$

$\square$

The implication of this result is that at least along the trajectories generated by the optimal policy, the representation will have monotonically decreasing and well-behaved embedding distances to the goal. Now, since in practice, VIP is trained on goal-directed (human video) trajectories, which are near-optimal for goal-reaching, we expect this smoothness result to be informative about VIP's embedding practical behavior and help formalize out intuition about the mechanism of implicit time contrastive learning. As confirmed by our qualitative study in Section H.4, We highlight that VIP's embedding is indeed much smoother than other baselines along test trajectories on both Ego4D and on our real-robot dataset. This smoothness along optimal paths makes it easier for the downstream control optimizer to discover these paths, conferring VIP representation effective zero-shot reward-specification capability that is not attained by any other comparison.

# F   VIP Training Details

## F.1   Dataset Processing and Sampling

We use the exact same pre-processed Ego4D dataset as in R3M, in which long raw videos are first processed into shorter videos consisting of 60-70 frames each. In total, there are approximately 72000 clips and 4.3 million frames in the dataset. Within a sampled batch, we first sample a set of videos, and then sample a sub-trajectory from each video (Step 3 in Algorithm 2). In this formulation, each sub-trajectory is treated as a video segment from the algorithm's perspective; this can viewed as a variant of trajectory data augmentation. As in R3M, we apply random crop at a video level within a batch, so all frames from the same video sub-trajectory are cropped the same way. Then, each raw observation is resized and center-cropped to have shape $224 \times 224 \times 3$ before passed into the visual encoder. Finally, as in standard contrastive learning and R3M, for each sampled sub-trajectory $\{o_t^i, ..., o_k^i, o_{k+1}^i, ..., o_T^i\}$, we also sample additional 3 negative samples $(\tilde{o}_j, \tilde{o}_{j+1})$ from separate video sequences to be included in the log-sum-exp term in $\mathcal{L}(\phi)$.

## F.2   VIP Hyperparameters

Hyperparameters used can be found in Table 2.

## F.3   VIP Pytorch Pseudocode

In this section, we present a pseudocode of VIP written in PyTorch (Paszke et al., 2019), Algorithm 3. As shown, the main training loop can be as short as 10 lines of code.

Table 2: VIP Architecture & Hyperparameters.

|  | Name | Value |
|---|---|---|
| Architecture | Visual Backbone | ResNet50 (He et al., 2016) |
|  | FC Layer Output Dim | 1024 |
| Hyperparameters | Optimizer | Adam (Kingma & Ba, 2014) |
|  | Learning rate | 0.0001 |
|  | $L_1$ weight penalty | 0.001 |
|  | $L_1$ weight penalty | 0.001 |
|  | Mini-batch size | 32 |
|  | Discount factor $\gamma$ | 0.98 |

---

**Algorithm 3** VIP PyTorch Pseudocode

---

```
# D: offline dataset
# phi: vision architecture

# training loop
for (o_0, o_t1,o_t2, g) in D:
    phi_g = phi(o_g)
    V_0 = - torch.linalg.norm(phi(o_0), phi_g)
    V_t1 = - torch.linalg.norm(phi(o_t1), phi_g)
    V_t2 = - torch.linalg.norm(phi(o_t2), phi_g)
    VIP_loss = (1-gamma)*-V_0.mean() + torch.logsumexp(V_t1+1-gamma*V_t2)
    optimizer.zero_grad()
    VIP_loss.backward()
    optimizer.step()
```

---

## G  Simulation Experiment Details.

### G.1  FrankaKitchen Task Descriptions

In this section, we describe the FrankaKitchen suite for our simulation experiments. We use 12 tasks from the v0.1 version[2] of the environment.

We use the environment default initial state as the initial state and frame for all tasks in the Hard setting. In the Easy setting, we use the 20th frame of a demonstration trajectory and its corresponding environment state as the initial frame and state. The goal frame for both settings is chosen to be the last frame of the same demonstration trajectory. The initial frames and goal frame for all 12 tasks and 3 camera views are illustrated in Figure 5-6. In the Easy setting, the horizon for all tasks is 50 steps; in the Hard setting, the horizon is 100 steps. Note that using the 20th frame as the initial state is a crude way for initializing the robot, and for some tasks, this initialization makes the task substantially easier, whereas for others, the task is still considerably difficult. Furthermore, some tasks become naturally more difficult depending on camera viewpoints. For these reasons, it is worth noting that our experiment's emphasis is on the *aggregate* behavior of pre-trained representations, instead of trying to solve any particular task as well as possible.

### G.2  In-Domain Representation Probing

In this section, we describe the experiment we performed to generate the in-domain VIP vs. TCN comparison in Figure 4. We fit VIP and TCN representations using 100 demonstrations from the FrankaKitchen sdoor_open task (center view). For TCN, we use R3M's implementation of the TCN loss without any modification; this also allows our findings in Figure 4 to extend to the main experiment section. The visual architecture is ResNet34, and the output dimension is 2, which enables us to directly visualize the learned embedding. Different from the out-of-domain version of VIP, we also do not perform weight penalty, trajectory-level random cropping data augmentation, or additional

---

[2]https://github.com/vikashplus/mj_envs/tree/v0.1real/mj_envs/envs/relay_kitchen

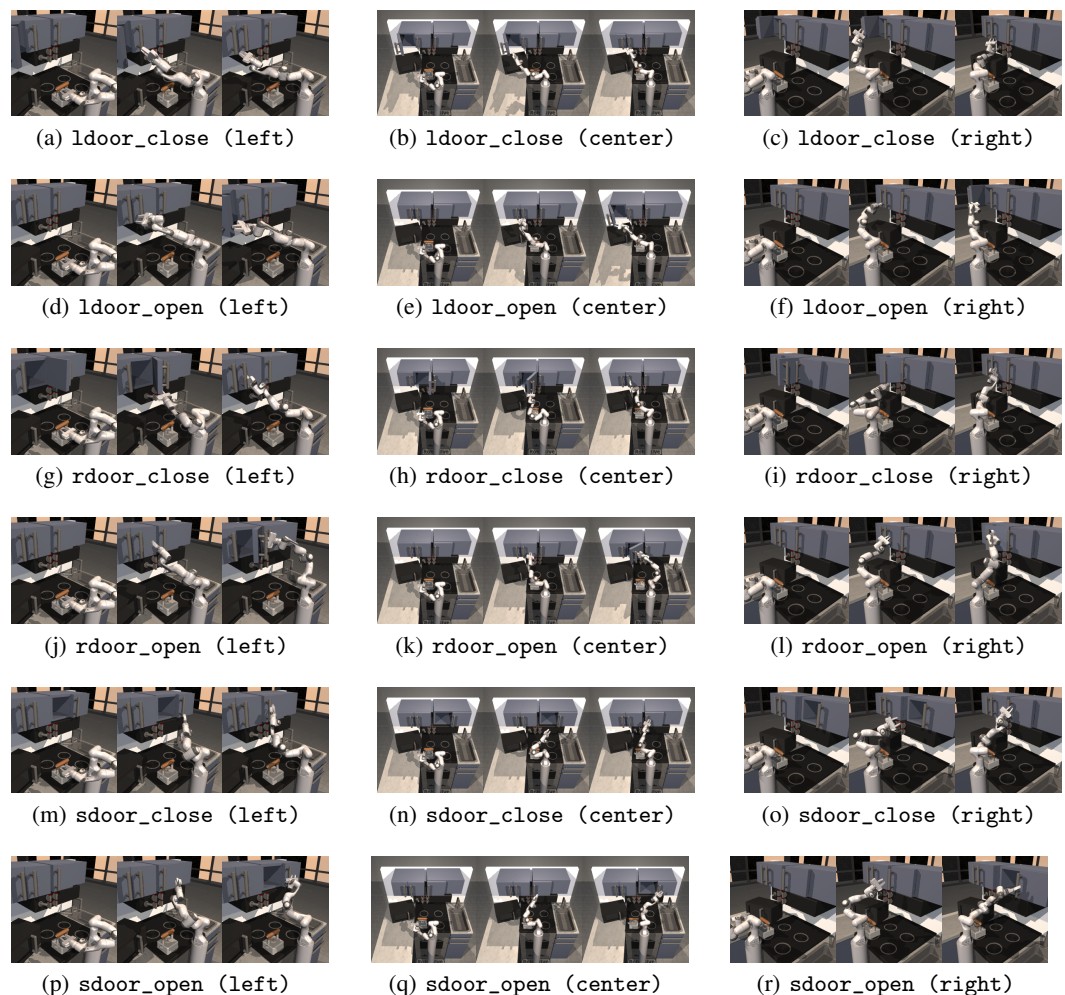

Figure 5: Initial frame (Easy), initial frame (Hard), and goal frame for all 12 tasks and 3 camera views in our FrankaKitchen suite.

negative sampling. Besides these choices, we use the same hyperparameters as in Table 2 and train for 2000 batches.

### G.3 Trajectory Optimization

We use a publicly available implementation of MPPI[3], and make no modification to the algorithm or the default hyperparameters. In particular, the planning horizon is 12 and 32 sequences of actions are proposed per action step. Because the embedding reward ((4)) is the goal-embedding distance difference, the score (i.e., sum of per-transition reward) of a proposed sequence of actions is equivalent to the negative embedding distance (i.e., $S_\phi(\phi(o_T); \phi(g))$) at the last observation.

### G.3.1 Robot and Object Pose Error Analysis

In this section, we visualize the per-step robot and object pose $L_2$ error with respect to the goal-image poses. We report the non-cumulative curves (on the success rate as well) for more informative analysis.

---

[3]https://github.com/aravindr93/trajopt/blob/master/trajopt/algos/mppi.py

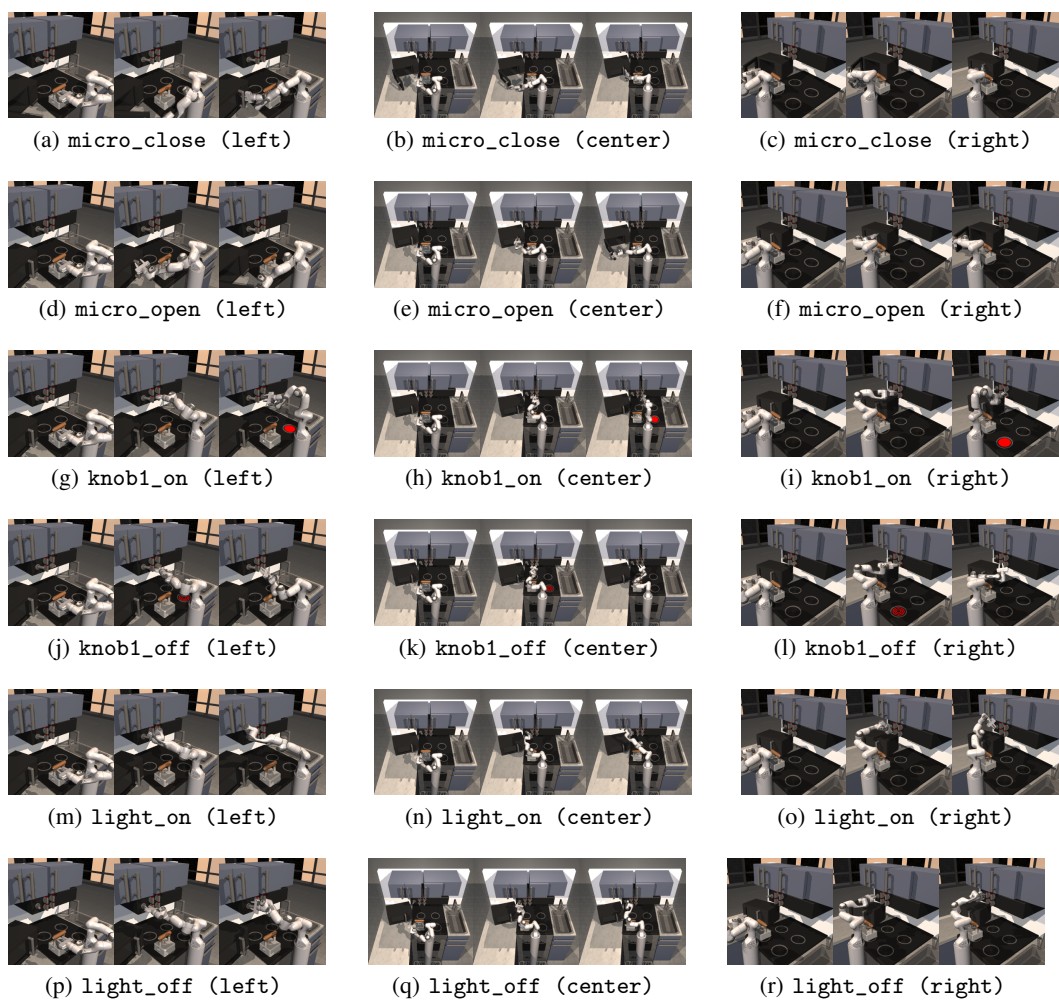

(a) `micro_close` (left)  (b) `micro_close` (center)  (c) `micro_close` (right)

(d) `micro_open` (left)  (e) `micro_open` (center)  (f) `micro_open` (right)

(g) `knob1_on` (left)  (h) `knob1_on` (center)  (i) `knob1_on` (right)

(j) `knob1_off` (left)  (k) `knob1_off` (center)  (l) `knob1_off` (right)

(m) `light_on` (left)  (n) `light_on` (center)  (o) `light_on` (right)

(p) `light_off` (left)  (q) `light_off` (center)  (r) `light_off` (right)

Figure 6: Initial frame (Easy), initial frame (Hard), and goal frame for all 12 tasks and 3 camera views in our FrankaKitchen suite.

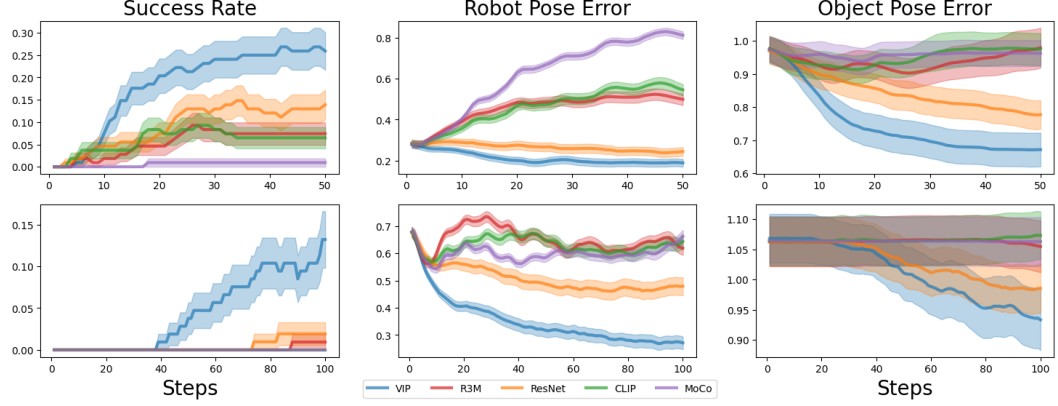

Figure 7: Trajectory optimization results with pose errors.

Table 3: Real-world robotics tasks descriptions.

| Environment | Object Type | Dataset | Success Criterion |
|---|---|---|---|
| CloseDrawer | Articulated Object | 10 demos + 20 failures | the drawer is closed enough that the spring loads. |
| PushBottle | Transparent Object | 20 demonstrations | the bottle is parallel to the goal line set by the icecream cone. |
| PlaceMelon | Soft Object | 20 demonstrations | the watermelon toy is fully placed in the plate. |
| FoldTowel | Deformable Object | 20 demonstrations | the bottom half of the towel is cleanly covered by the top half. |

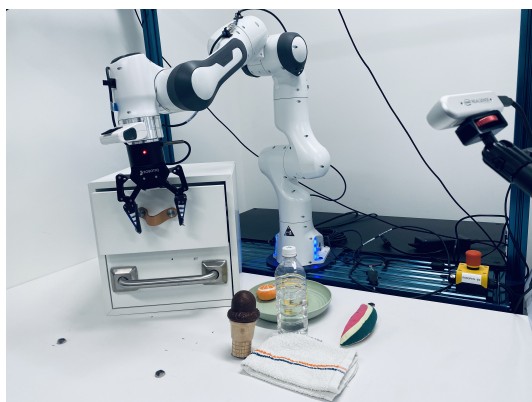

Figure 8: Real-robot setup.

## G.4 Reinforcement Learning

We use a publicly available implementation of NPG[4], and make no modification to the algorithm or the default hyperparameters. In the Easy (resp. Hard) setting, we train the policy until 500000 (resp. 1M) real environment steps are taken. For evaluation, we report the cumulative maximum success rate on 50 test rollouts from each task configuration (50*108=5400 total rollouts) every 10000 step.

# H Real-World Robot Experiment Details

## H.1 Task Descriptions

The robot learning environment is illustrated in Figure 8; a RealSense camera is mounted on the right edge of the table, and we only use the RGB image stream without depth information for data collection and policy learning.

We collect offline data $D_{\text{task}}$ for each task via kinesthetic playback, and the object initial placement is randomized for each trajectory. On the simplest CloseDrawer task, we combine 10 expert demonstrations with 20 sub-optimal failure trajectories to increase learning difficulty. For the other three tasks, we collect 20 expert demonstrations, which we found are difficult enough for learning good policies. Each demonstration is 50-step long collected at 25Hz. The initial state for the robot is fixed for each demonstration and test rollout, but the object initial position is randomized. The task success is determined based on a visual criterion that we manually check for each test rollout. The full task breakdown is described in Table 3.

Each task is specified via a set of goal images that are chosen to be the last frame of all demonstrations for the task. Hence, the goal embedding used to compute the embedding reward ((4)( for each task is the average over the embeddings of all goal frames.

The tasks (in their initial positions) using a separate high-resolution phone camera are visualized in Figure 9. Sample demonstrations in the robot camera view are visualized in Figure 10.

---

[4]https://github.com/aravindr93/mjrl/blob/master/mjrl/algos/npg_cg.py

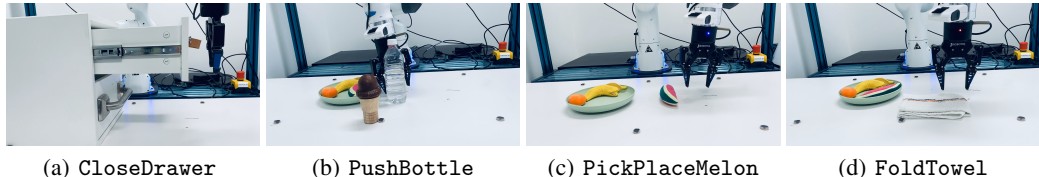

| (a) CloseDrawer | (b) PushBottle | (c) PickPlaceMelon | (d) FoldTowel |

Figure 9: Side-view of real-robot tasks using a high-resolution smartphone camera.

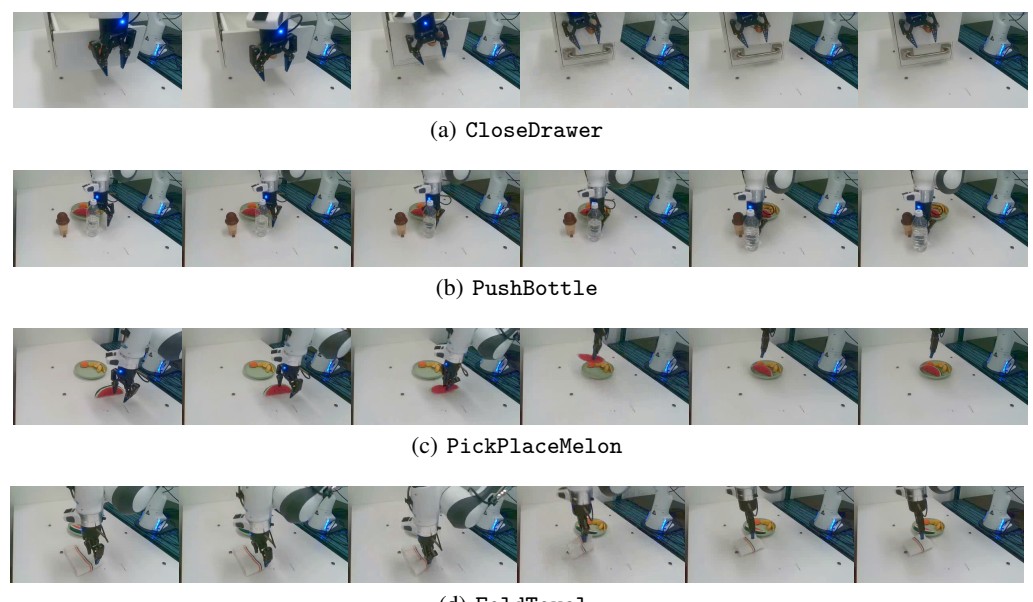

(a) CloseDrawer

(b) PushBottle

(c) PickPlaceMelon

(d) FoldTowel

Figure 10: Real-robot task demonstrations (every 10th frame) in robot camera view. The first and last frames in each row are representative of initial and final goal observaions for the respective task.

## H.2   Training and Evaluation Details

The policy network is implemented as a 2-layer MLP with hidden sizes $[256, 256]$. As in R3M's real-world robot experiment setup, the policy takes in concatenated visual embedding of current observation and robot's proprioceptive state and outputs robot action. The policy is trained with a learning rate of 0.001, and a batch size of 32 for 20000 steps.

For RWR's temperature scale, we use $\tau = 0.1$ for all tasks, except CloseDrawer where we find $\tau = 1$ more effective for both VIP and R3M.

For policy evaluation, we use 10 test rollouts with objects randomly initialized to reflect the object distribution in the expert demonstrations. The rollout horizon is 100 steps.

## H.3   Additional Analysis & Context

**Offline RL vs. imitation learning for real-world robot learning.** Offline RL, though known as the data-driven paradigm of RL (Levine et al., 2020), is not necessarily data *efficient* (Agarwal et al., 2021), requiring hundreds of thousands of samples even in low-dimensional simulated tasks, and requires a dense reward to operate most effectively (Mandlekar et al., 2021; Yu et al., 2022). Furthermore, offline RL algorithms are significantly more difficult to implement and tune compared to BC (Kumar et al., 2021; Zhang & Jiang, 2021). As such, the dominant paradigm of real-world robot learning is still learning from demonstrations (Jang et al., 2022; Mandlekar et al., 2018; Ebert et al.,

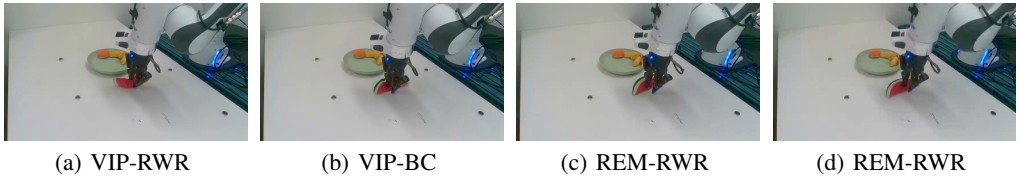

| (a) VIP-RWR | (b) VIP-BC | (c) REM-RWR | (d) REM-RWR |

Figure 11: Comparison of failure trajectories on `PickPlaceMelon`. VIP-RWR is still able to reach the critical state of gripping watermelon, whereas baselines fail.

2021). With the advent of VIP-RWR, offline RL may finally be a practical approach for real-world robot learning at scale.

**Performance of R3M-BC.** Our R3M-BC, though able to solve some of the simpler tasks, appears to perform relatively worse than the original R3M-BC in Nair et al. (2022) on their real-world tasks. To account for this discrepancy, we note that our real-world experiment uses different software-hardware stacks and tasks from the original R3M real-world experiments, so the results are not directly comparable. For instance, camera placement, an important variable for real-world robot learning, is chosen differently in our experiment and that of R3M; in R3M, a different camera angle is selected for each task, whereas in our setup, the same camera view is used for all tasks. Furthermore, we emphasize that our focus is not the absolute performance of R3M-BC, but rather the relative improvement R3M-RWR provides on top of R3M-BC.

### H.4 Qualitative Analysis

In this section, we study several interesting policy behaviors VIP-RWR acquire. Policy videos are included in our supplementary video.

**Robust key action execution.** VIP-RWR is able to execute key actions more robustly than the baselines; this suggests that its reward information helps it identify necessary actions. For example, as shown in Figure 11, on the `PickPlaceMelon` task, failed VIP-RWR rollouts at least have the gripper grasp onto the watermelon, whereas for other baselines, the failed rollouts do not have the watermelon between the gripper and often incorrectly push the watermelon to touch the plate's outer edge, preventing pick-and-place behavior from being executed.

**Task re-attempt.** We observe that VIP-RWR often learns more robust policies that are able to perform recovery actions when the task is not solved on the first attempt. For instance, in both `CloseDrawer` and `FoldTowel`, there are trials where VIP-RWR fails to close the drawer all the way or pick up the towel edge right away; in either case, VIP-RWR is able to re-attempt and solves the task (see our supplementary video). This is a known advantage of offline RL over BC (Kumar et al., 2022; Levine et al., 2020); however, we only observe this behavior in VIP-RWR and not R3M-RWR, indicating that this advantage of offline RL is only realized when the reward information is sufficiently informative.

## I Additional Results

### I.1 Value-Based Pre-Training Ablation: Least-Square Temporal-Difference

While VIP is the first value-based pre-training approach and significantly outperforms all existing methods, we show that this effectiveness is also unique to VIP and not to training a value function. To this end, we show that a simpler value-based baseline does not perform as well. In particular, we consider Least-Square Temporal-Difference policy *evaluation* (**LSTD**) (Bradtke & Barto, 1996; Sutton & Barto, 2018) to assess the importance of the choice of value-training objective:

$$\min_{\phi} \mathbb{E}_{(o,o',g) \sim D} \left[ \left( \tilde{\delta}_g(o) + \gamma V(\phi(o'); \phi(g)) - V(\phi(s), \phi(g)) \right)^2 \right], \tag{31}$$

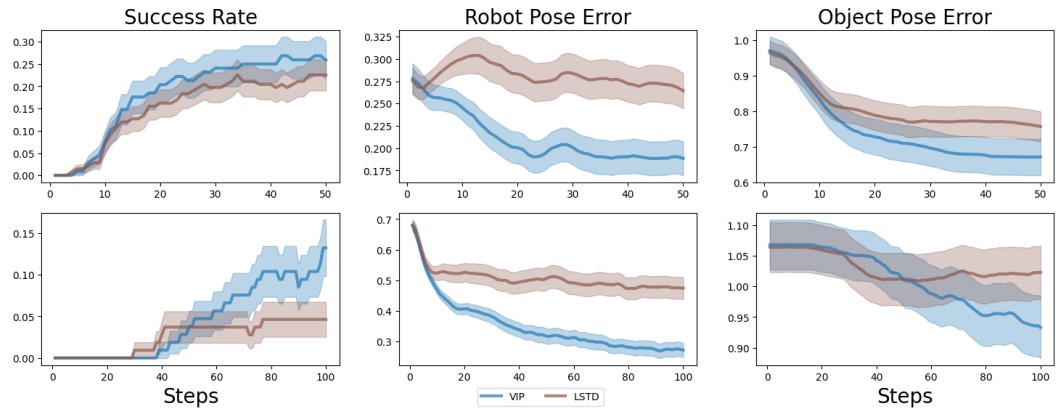

Figure 12: VIP vs. LSTD Trajectory Optimization Comparison.

Table 4: Visual Imitation Learning Results.

|  | *Self-Supervised* | | | | *Supervised* | |
|  | VIP (E) | LSTD (E) | R3M-Lang (E) | MOCO (I) | R3M (E) | ResNet50 (I) | CLIP (Internet) |
| --- | --- | --- | --- | --- | --- | --- | --- |
| Success Rate | **53.6** | 51.5 | 51.2 | 45.0 | **55.9** | 41.8 | 44.3 |

in which we also parameterize $V$ as the negative $L_2$ embedding distance as in VIP. Given that human videos are reasonably goal-directed, the value of the human behavioral policy computed via LSTD should be a decent choice of reward; however, LSTD does not capture the long-range dependency of initial to goal frames (first term in (12)), nor can it obtain a value function that outperforms that of the behavioral policy. We train LSTD using the exact same setup as in VIP, differing in only the training objective, and compare it against VIP in our trajectory optimization settings.

As shown in Fig. 12, interestingly, LSTD already works better than all prior baselines in the Easy setting, indicating that value-based pre-training is indeed favorable for reward-specification. However, its inability to capture long range temporal dependency as in VIP (the first term in VIP's objective) makes it far less effective on the Hard setting, which require extended smoothness in the reward landscape to solve given the distance between the initial observation and the goal. These results show that VIP's superior reward specification comes precisely from its ability to capture both long-range temporal dependencies and local temporal smoothness, two innate properties of its dual value objective and the associated implicit time contrastive learning interpretation. To corroborate these findings, we have also included LSTD in our qualitative reward curve and histogram analysis in App. I.4, I.6, and I.7 and finds that VIP generates much smoother embedding than LSTD.

## I.2 Visual Imitation Learning

One alternative hypothesis to VIP's smoother embedding for its superior reward-specification capability is that it learns a better visual representation, which then naturally enables a better visual reward function. To investigate this hypothesis, we compare representations' capability as a pure visual encoder in a visual imitation learning setup. We follow the training and evaluation protocol of (Nair et al., 2022) and consider 12 tasks combined from FrankaKitchen, MetaWorld (Yu et al., 2020), and Adroit (Rajeswaran et al., 2017), 3 camera views for each task, and 3 demonstration dataset sizes, and report the aggregate average maximum success rate achieved during training. **R3M-Lang** is the publicly released R3M variant without supervised language training. The average success rates over all tasks are shown in Table 4; the letter inside () stands for the pre-training dataset with $E$ referring to Ego4D and $I$ Imagenet.

These results suggest that with current pre-training methods, the performance on visual imitation learning may largely be a function of the pre-training dataset, as all methods trained on Ego4D, even our simple baseline LSTD, performs comparably and are much better than the next best baseline

not trained on Ego4D. Conversely, this result also suggests that despite not being designed for this purely supervised learning setting, value-based approaches constitute a strong baseline, and VIP is in fact currently the state-of-art for self-supervised methods. While these results highlight that VIP is effective even as a pure visual encoder, a necessary requirement for joint effectiveness for visual reward and representation, it fails to explain why VIP is far superior to R3M in reward-based policy learning. As such, we conclude that studying representations' capability as a pure visual encoder may not be sufficient for distinguishing representations that can additionally perform zero-shot reward-specification.

### I.3 Embedding and True Rewards Correlation

In this section, we create scatterplots of embedding reward vs. true reward on the trajectories MPPI have generated to assess whether the embedding reward is correlated with the ground-truth dense reward. More specifically, for each transition in the MPPI trajectories in Figure 2, we plot its reward under the representation that was used to compute the reward for MPPI versus the true human-crafted reward computed using ground-truth state information. The dense reward in FrankaKitchen tasks is a weighted sum of (1) the negative object pose error, (2) the negative robot pose error, (3) bonus for robot approaching the object, and (4) bonus for object pose error being small. This dense reward is highly tuned and captures human intuition for how these tasks ought to be best solved. As such, high correlation indicates that the embedding is able to capture both intuitive robot-centric and object-centric task progress from visual observations. We only compare VIP and R3M here as a proxy for comparing our implicit time contrastive mechanism to the standard time contrastive learning.

The scatterplots over all tasks and camera views (Easy setting) are shown in Figure 13,14, and 15. VIP rewards exhibit much greater correlation with the ground-truth reward on its trajectories that do accomplish task, indicating that when VIP does solve a task, it is solving the task in a way that matches *human* intuition. This is made possible via large-scale value pre-training on diverse human videos, which enables VIP to extract a human notion of task-progress that transfers to robot tasks and domains. These results also suggest that VIP has the potential of *replacing* manual reward engineering, providing a data-driven solution to the grand challenge of reward engineering for manipulation tasks. However, VIP is not yet perfect in its current form. Both methods exhibit local minima where high embedding distances in fact map to lower true rewards; however, this phenomenon is much severe for R3M. On 8 out of 12 tasks, VIP at least has one camera view in which its rewards are highly correlated with the ground-truth rewards on its MPPI trajectories.

### I.4 Embedding Distance Curves

In Figure 16, we present additional embedding distance curves for all methods on Ego4D and our real-robot offline RL datasets. For Ego4D, we randomly sample 4 videos of 50-frame long (see Appendix I.5 for how these short snippets are sampled), and for our robot dataset, we compute the embedding distance curves for the 4 sample demonstrations in Figure 10. As shown, on all tasks in the real-robot dataset, VIP is distinctively more smooth than any other representation. This pattern is less accentuated on Ego4D. This is because a randomly sampled 50-frame snippet from Ego4D may not coherently represent a task solved from beginning to completion, so an embedding distance curve is not inherently supposed to be smoothly declining. Nevertheless, VIP still exhibits more local smoothness in the embedding distance curves, and for the snippets that do solve a task (the first two videos), it stands out as the smoothest representation.

### I.5 Embedding Distance Curve Bumps

In this section, we compute the fraction of negative embedding rewards (equivalently, positive slopes in embedding embedding distance curves) for each video sequence and average over all video sequences in a dataset. Each sequence in our robot dataset is of 50 frames, and we use each sequence without any further truncation. For Ego4D, video sequences are of variable length. For each long sequence of more than 50 frames, we use the first 50 frames. We do not include videos shorter than

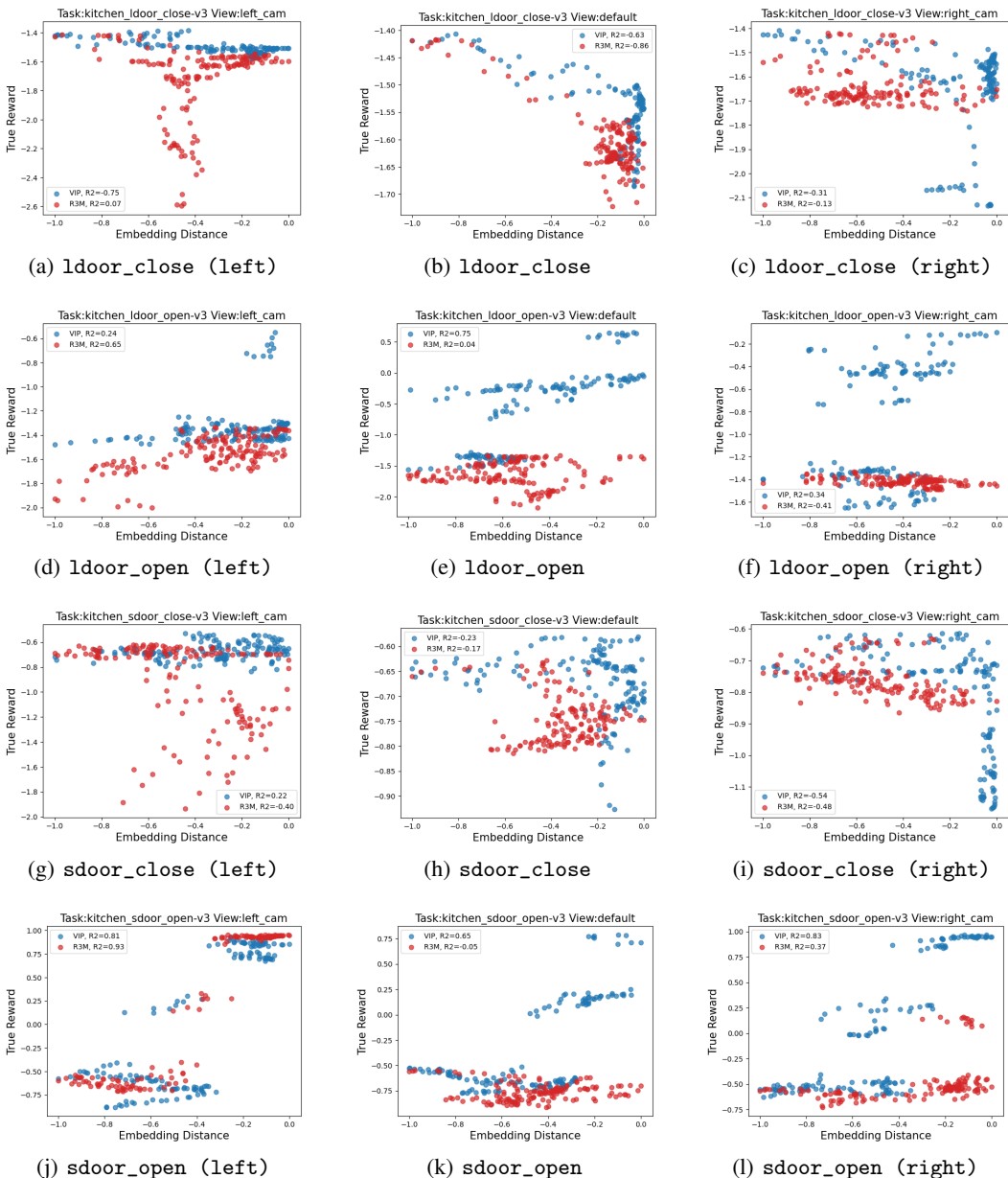

Figure 13: Embedding reward vs. ground-truth human-engineered reward correlation (VIP vs. R3M) part 1.

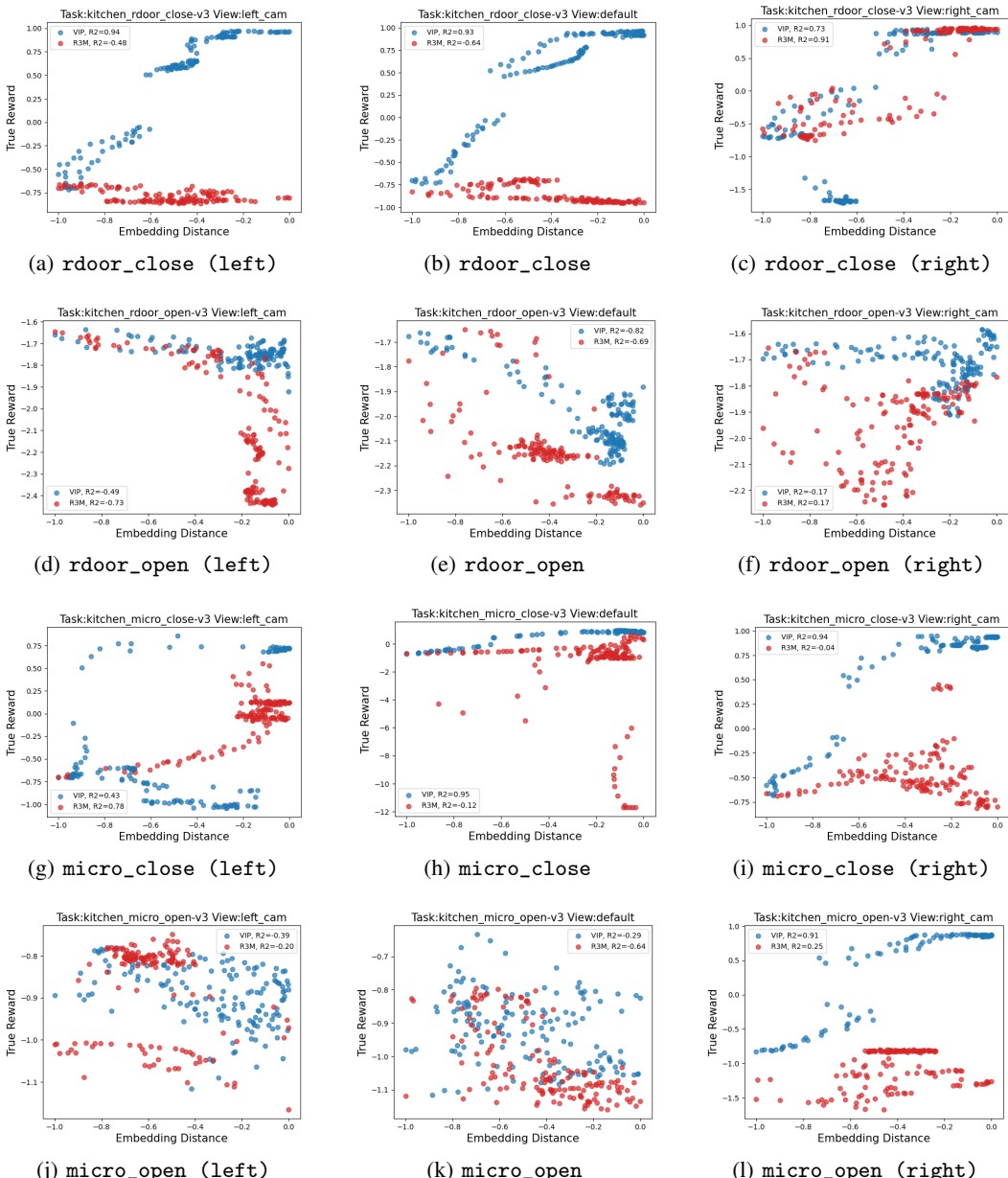

Figure 14: Embedding reward vs. ground-truth human-engineered reward correlation (VIP vs. R3M) part 2.

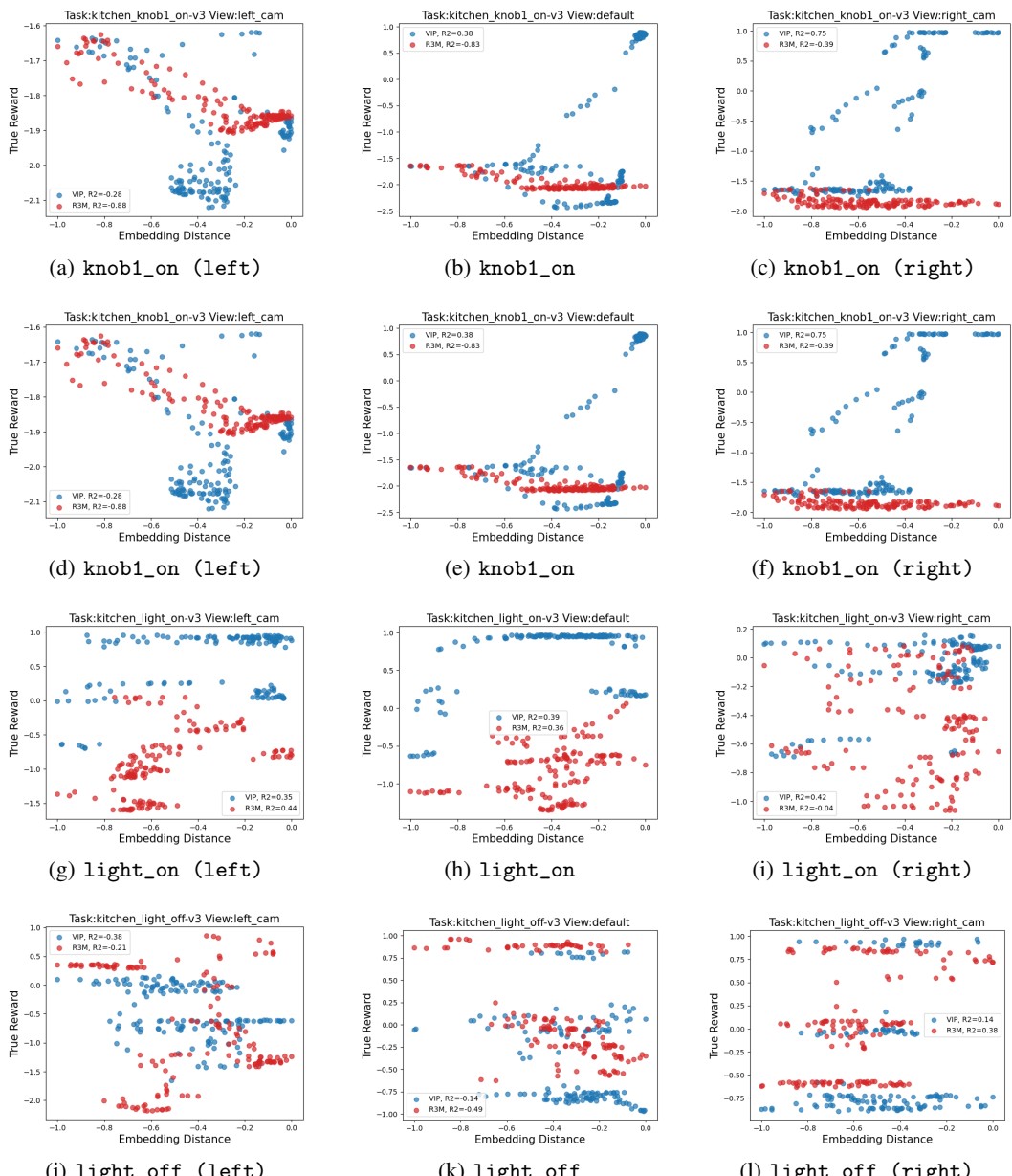

Figure 15: Embedding reward vs. ground-truth human-engineered reward correlation (VIP vs. R3M) part 3.

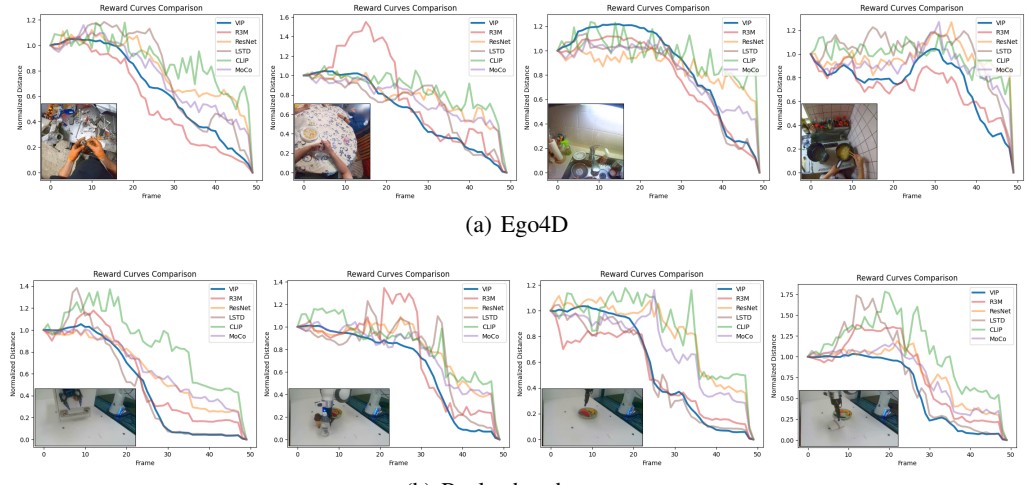

(a) Ego4D

(b) Real-robot dataset

Figure 16: Additional embedding distance curves on Ego4D and real-robot videos.

Table 5: Proportion of bumps in embedding distance curves.

| Dataset | VIP (Ours) | R3M | ResNet50 | MOCO | CLIP |
|---|---|---|---|---|---|
| Ego4D | **0.253** $\pm$ 0.117 | 0.309 $\pm$ 0.097 | 0.414 $\pm$ 0.052 | 0.398 $\pm$ 0.057 | 0.444 $\pm$ 0.047 |
| In-House Robot Dataset | **0.243** $\pm$ 0.066 | 0.323 $\pm$ 0.076 | 0.366 $\pm$ 0.046 | 0.380 $\pm$ 0.052 | 0.438 $\pm$ 0.046 |

50 frames, in order to make the average fraction for each representation comparable between the two distinct datasets. Note that for Ego4D, due to its in-the-wild nature, it is not guaranteed that a 50-frame segment represents one task being solved from beginning to completion, so there may be naturally bumps in the embedding distance curve computed with respect to the last frame, as earlier frames may not actually be progressing towards the last frame in a goal-directed manner.The full results are shown in Table 5. VIP has fewest bumps in Ego4D videos, and this notion of smoothness transfer to the robot dataset. Furthermore, since the robot videos are in fact visually simpler and each video is guaranteed to be solving one task, the bump rate is actually *lower* despite the domain gap. While this observation generally also holds true for other representations, it notably does not hold for R3M, which is trained using standard time contrastive learning.

## I.6  Embedding Reward Histograms (Real-Robot Dataset)

We present the reward histogram comparison against all baselines in Figure 17. The trend of VIP having more small, positive rewards and fewer extreme rewards in either direction is consistent across all comparisons.

## I.7  Embedding Reward Histograms (Ego4D)

We present the reward histogram comparison against all baselines in Figure 18. The histograms are computed using the same set of 50-frame Ego4D video snippets as in Appendix I.5. The y-axis is in log-scale due to the large total count of Ego4D frames. As discussed, Ego4D video segments are less regular than those in our real-robot dataset, and this irregularity contributes to all representations having significantly more negative rewards compared to their histograms on the real-robot dataset. Nevertheless, the relative difference ratio's pattern is consistent, showing VIP having far more rewards that lie in the first positive bin. Furthermore, VIP also has significantly fewer extreme negative rewards compared to all baselines.

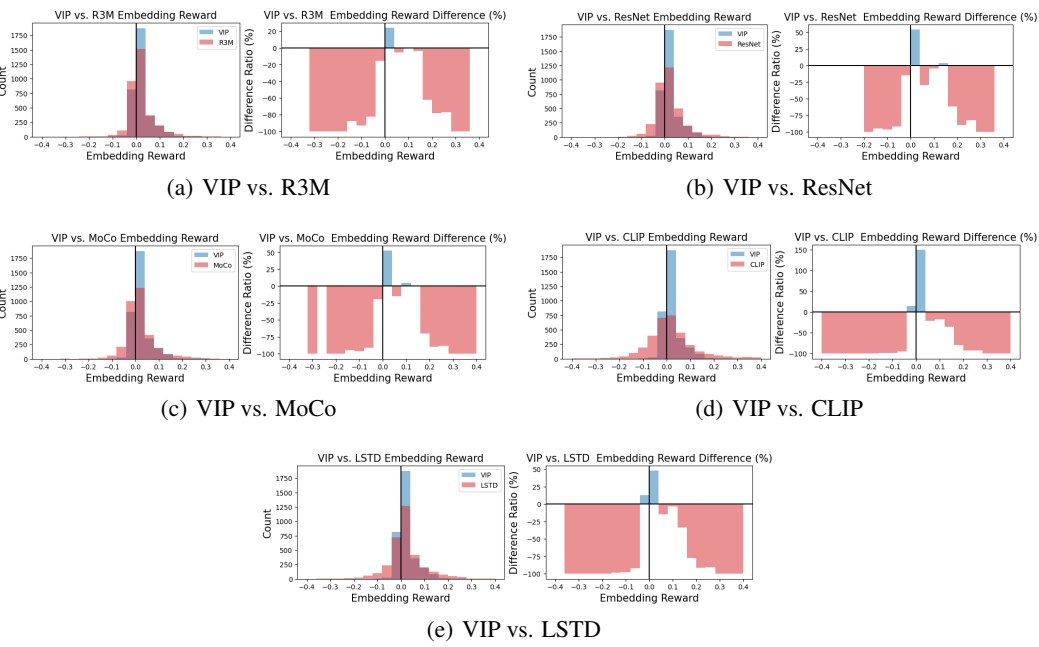

Figure 17: Embedding reward histogram comparison on real-robot dataset.

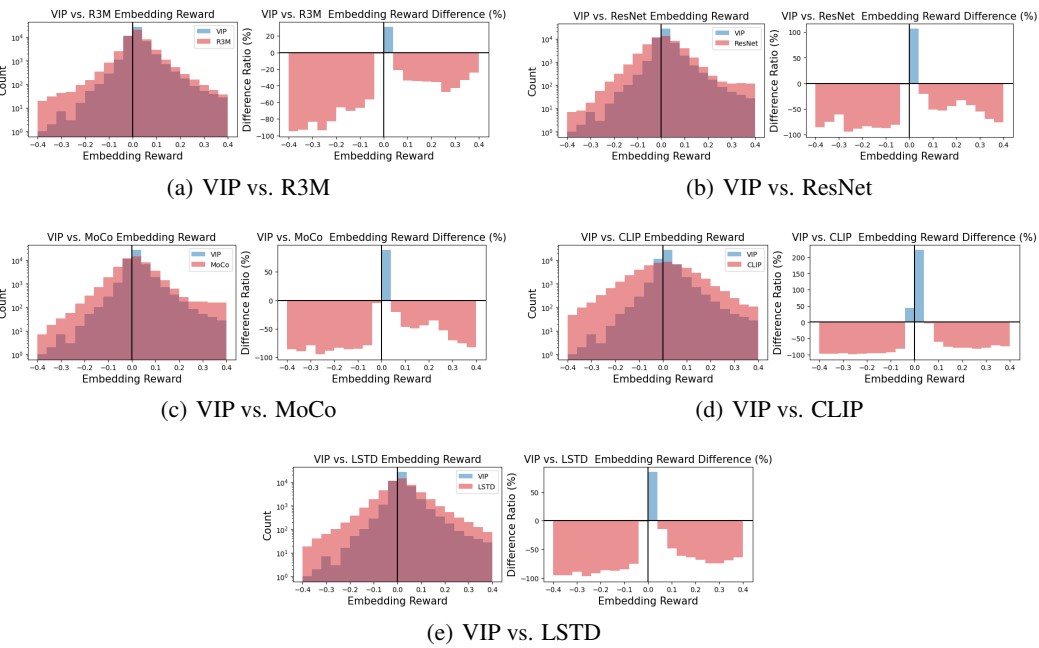

Figure 18: Embedding reward histogram comparison on Ego4D videos.

