# OpenReview forum: "Towards Universal Visual Reward and Representation via Value-Implicit Pre-Training"
_NeurIPS.cc/2022/Workshop/Offline_RL — Offline RL Workshop NeurIPS 2022_

### Official Review · Reviewer_AsZp · 2022-10-10
**An effective approach with theoretical foundation**

**Rating:** 7
**Confidence:** 4

**Review:**

Summary:

This paper considers learning pre-trained visual representations for robotics tasks. The authors derive a dual goal-conditioned value function objective that does not depend on actions. Given this objective, the visual representation can be learned from human videos without action labels, and the well-trained representation can be used to provide a dense reward function for downstream robotics tasks. Experiments on the simulated platform and real robot demonstrate that the proposed pre-trained representation improves policy learning algorithms more than other existing pre-trained representations.

Strength:

The proposed objective function is grounded on the theoretical foundation of a dual problem, and it is proven to be equivalent to implicit time contrastive learning.

The proposed method can apply to any human videos, without access to action labeling. This makes it possible to leverage large amounts of cheap data online.

The evaluation is performed not only on a simulation platform but also on a real robot.

Weakness:

Since the proposed method is equivalent to implicit time contrastive learning, a natural question is whether the naive TCN can reach a good performance similar to the proposed method. If not, why? what's the subtle advantage of VIP in comparison with TCN?

Are there any requirements about the human video dataset for pre-training? To make the learned representation beneficial, how should the human video dataset relate to the downstream tasks? If the video dataset is not appropriate, is it possible that the pre-trained representation even hurts policy learning?

Could the learned frozen representation be directly used as input into the policy networks for policy learning? Will this help improve the performance? If not, why?

The proposed method with the theoretical ground is simple and effective based on the experimental results, though there are still some questions for the evaluation and analysis part. Overall, I vote for acceptance.

---

### Official Review · Reviewer_3xUx · 2022-10-19
**Offline pretraining of implicit temporal distance based value functions on human videos can be used for few-shot real world RL. Clear accept.**

**Rating:** 8
**Confidence:** 3

**Review:**

They leverage the fact that the Fenchel dual of a KL-regularized offline RL objective does not make use of actions to pretrain on a large-scale offline human dataset (Ego4D) without action labels. By training on videos labelled with a trivial reward function where all non-goal states are assigned a -1 reward and the goal 0, the value function so learned corresponds to the discounted temporal distance to the goal. They show that this can be seen as a novel implicit time contrastive learning method, where the goal conditioned value behaves as a similarity metric. They use this method to pretrain on offline human videos and show that this can improve trajectory optimization using MPPI and online RL significantly (considering the value smoothly increases as the goal approaches). Further they also demonstrate good results on real world few-shot RL tasks.

Significance: How offline human videos can be used to produce a smooth reward is certainly interesting and the experiments and arguments are compelling and significant.

Novelty: Similar to prior work on contrastive learning as GCRL but applied in the Offline RL setting with human videos.

Impact: Provides a valuable link between Offline RL and time contrastive learning and shows how this can be used to obtain good results in the few shot real world RL tasks.

Clarity: Quite clear, although due to the page limit most of the content is in the appendix. Overall the results could have been much better summarized within the page limit. Would be good to have a short section on the comparisons with related work in the main paper rather than the appendix.

Overall significantly above acceptance threshold as the experiments are compelling. But comparison with baselines is limited and the content can be much better organized and summarized in the main paper.